# Identification and Validation of the Anoikis-Related Gene Signature as a Novel Prognostic Model for Cervical Squamous Cell Carcinoma, Endocervical Adenocarcinoma, and Revelation Immune Infiltration

**DOI:** 10.3390/medicina59020358

**Published:** 2023-02-13

**Authors:** Qin-Qin Jin, Jie Mei, Lin Hong, Rui Wang, Shuang-Yue Wu, Sen-Lin Wang, Xi-Ya Jiang, Yin-Ting Yang, Hui Yao, Wei-Yu Zhang, Yu-Ting Zhu, Jie Ying, Lu Tian, Guo Chen, Shu-Guang Zhou

**Affiliations:** 1Department of Gynecology, Maternal and Child Medical Centre of Anhui Medical University, Hefei 230001, China; 2Department of Gynecology, Anhui Province Maternity and Child Healthcare Hospital, Hefei 230001, China; 3Office of Health Care, Hefei Municipal Health Commission, Hefei 230071, China; 4Department of Clinical Laboratory, Anhui Province Maternity and Child Healthcare Hospital, Hefei 230001, China

**Keywords:** anoikis, cervical squamous cell carcinoma, endocervical adenocarcinoma, prognosis, therapeutic target

## Abstract

*Background and Objectives*: Cervical squamous cell carcinoma and endocervical adenocarcinoma (CESC) are malignant disorders with adverse prognoses for advanced patients. Anoikis, which is involved in tumor metastasis, facilitates the survival and separation of tumor cells from their initial site. Unfortunately, it is rarely studied, and in the literature, studies have only addressed the prognosis character of anoikis for patients with CESC. *Materials and Methods*: We utilized anoikis-related genes (ANRGs) to construct a prognostic signature in CESC patients that were selected from the Genecards and Harmonizome portals. Furthermore, we revealed the underlying clinical value of this signature for clinical maneuvers by providing clinical specialists with an innovative nomogram on the basis of ANRGs. Finally, we investigated the immune microenvironment and drug sensitivity in different risk groups. *Results*: We screened six genes from fifty-eight anoikis-related differentially expressed genes in the TCGA-CESC cohort, and we constructed a prognostic signature. Then, we built a nomogram combined with CESC clinicopathological traits and risk scores, which demonstrated that this model may improve the prognosis of CESC patients in clinical therapy. Next, the prognostic risk scores were confirmed to be an independent prognostic indicator. Additionally, we programmed a series of analyses, which included immune infiltration analysis, therapy-related analysis, and GSVA enrichment analysis, to identify the functions and mechanisms of the prognostic models during the progression of cancer in CESC patients. Finally, we performed quantitative reverse transcription polymerase chain reaction (qRT-PCR) to verify the six ANRGs. *Conclusions*: The present discovery verified that the predictive 6-anoikis-related gene (6-ANRG) signature and nomogram serve as imperative factors that might notably impact a CESC patient’s prognosis, and they may be able to provide new clinical evidence to assume the role of underlying biological biomarkers and thus become indispensable indicators for prospective diagnoses and advancing therapy.

## 1. Introduction

Cervical cancer is the second most common cause of cancer-related mortality in women worldwide [1,2,3,4]. According to the available data, approximately 500,000 women are diagnosed with cervical squamous cell carcinoma and endocervical adenocarcinoma (CESC), and the disorder causes more than 300,000 fatalities worldwide annually [5,6,7]. One of the principal reasons for cervical cancer is driven by human papillomavirus (HPV) infection, which causes a high level of risk, despite the fact that HPV infection alone does not cause cervical cancer [8,9,10,11]. The overall survival (OS) rate is approximately 70% five years after pharmacotherapy for locally advanced cervical cancer [12,13]. Meanwhile, CESC is still the most typically seen type of cervical cancer. Currently, notwithstanding various studies that have intimated that aberrantly expressed tumor markers may be associated with cancer initiation, as well as progression and that tremendous endeavors have been made to explore novel biomarkers, cervical cancer remains refractory. Consequently, it is inevitable that we make a plea for the discovery of new prognostic biomarkers so as to enhance prognosis, decrease lethality, and produce rewarding treatment indications for cervical cancer patients.

Anoikis is a special apoptotic that is a consequence of the loss of or improper cell adhesion, which is paramount for the survival of tumor cells following separation from the extracellular matrix (ECM) [14,15,16]. Previous studies have found that an anti-anoikis mechanism in cancer cells is pivotal to cancer progression [17,18]. In recent years, although a multitude of mechanisms of anoikis in tumors have been proposed and validated, the role of prognostic molecules of anoikis-related genes underlying tumors remains to be defined. Meanwhile, few—or no—studies have paid attention to the connection between the anoikis process and distant metastasis in CESC.

Despite years of thorough examinations and continuous efforts, the emphasized cellular, as well as molecular, mechanisms remain intangible, and the clinical approaches for treating CESC are limited, which renders CESC a continued significant clinical challenge. Hence, in this study, we strove to investigate the predictive benefit of anoikis-related genes (ANRGs) in CESC patients and develop a prognostic signature based on the 6-anoikis-related gene (6-ANRG). Subsequently, we studied the discrepancies in a tumor-immune microenvironment in CESC patients based on their risk scores. Moreover, the present study on ANRGs in CESC patients can provide a foundation for the development of tumor pharmacotherapeutic targets, which is of great value for enhancing CESC patients’ quality of life.

## 2. Materials and Methods

### 2.1. Data Collection and Description

The RNA-seq and microarray profiles were obtained from The Cancer Genome Atlas (TCGA, https://portal.gdc.cancer.gov (accessed on 29 September 2022)) [19] and the Genotype-Tissue Expression (GTEx, http://xena.ucsc.edu/ (accessed on 29 September 2022)) portals [20], and they included 306 CESC tumor cases and 13 healthy controls. Furthermore, we obtained the GSE44001 cohort from the Gene Expression Omnibus (GEO, https://www.ncbi.nlm.nih.gov/geo/ (accessed on 29 September 2022)) database [21]. In total, 126 ANRGs were filtered from the Harmonizome (https://maayanlab.cloud/Harmonizome/ (accessed on 29 September 2022)) [22] and the GeneCard (https://www.genecards.org/ (accessed on 29 September 2022)) databases [23] (Appendix A). *p*-values of < 0.05 and a |log FoldChange| (|logFC|) of > 1 were regarded as the cut-off criteria. The “limma” R package was utilized to compare the differential expressions of 126 ANRGs between the tumor samples and the matched cervix controls in the TCGA-CESC dataset, and then 58 differentially expressed ANRGs were determined.

### 2.2. Functional Annotation and Protein–Protein Interaction (PPI) Analyses

For the purpose of exploring the biological functions of these genes, we carried out Gene Ontology (GO) and Kyoto Encyclopedia of Genes and Genomes (KEGG) analyses using the “enrichplot” R package. We chose the “Clusterprofiler” R package to carry out the Gene Set Enrichment Analysis (GSEA) to quantify the gene set activity. Next, we input these genes into the String (https://string-db.org/ (accessed on 29 September 2022) database to eliminate the genes with little connectivity, and then we used Cytoscape software (National Institute of General Medical Sciences (NIGMS), NY, U.S. (Version 3.7.1)) to visualize the results.

### 2.3. Development and Validation of Prognostic Signatures Based on ANRGs

Univariate Cox regression analysis was applied to filter the survival-related genes. Then, a least absolute shrinkage and selection operator (LASSO) regression analysis was performed using the “glmnet” R package. Next, multivariate Cox regression analysis was applied to determine the core genes, as well as to calculate the corresponding coefficients. According to the best λ values combined with corresponding coefficients, we selected six ANRGs to construct the risk model. The GSE44001 dataset served as an internal validation and was applied in the equivalent analysis approach to validate the availability of the prognosis model we constructed on the basis of the ANRGs of CESC. For each patient, the risk scores were calculated shown below. In the equation, βi and Expi represent the risk coefficient and the expression level of each gene, respectively. Both Kaplan–Meier (K-M) curve and time-dependent receiver operational feature curve (ROC) analyses were applied to estimate the accuracy of this prognosis signature.
$$risk\;scores=\sum_{}^{}\left(\beta i\times Expi\;\right)$$

### 2.4. Principal Component Analysis (PCA)

PCA serves as a serviceable tool that is broadly applied in decreasing dimensionality and extracting traits in the computer vision area. Here, the “scatterplot3d” R package was utilized to assess the differences in the two risk groups.

### 2.5. Establishment and Estimation of the Nomogram

The nomogram was established by integrating both the clinicopathological features and the risk scores. As an internal validation, a calibration plot was utilized to test the precision of the nomogram. In addition, the clinicopathological characteristics data and the risk scores were included in the multivariate Cox regression method to further verify that they could serve as independent prognosis factors.

### 2.6. Relationship between Risk Scores and Immune Cell Infiltration

The relative ratio of immune cells was quantified by the CIBERSORT and ssGSEA R scripts. Per sample, the sum total of all evaluated immune cell type scores was equal to one. Finally, the relationship between the immune infiltrating cells and the risk score values was examined using a Spearman correlation analysis.

### 2.7. Drug Sensitivity Analysis

The Genomics of Drug Sensitivity in Cancer (GDSC, https://www.cancerrxgene.org/(accessed on 30 September 2022) [24] portal was utilized to obtain the antitumor drug profiles. A Spearman correlation analysis was performed to calculate the correlation between drug sensitivity and the risk scores. The “pRRophetic” R package was used to calculate the half-maximum inhibitory concentration (IC 50) for comparing the drug sensitivity.

### 2.8. Gene Set Variation Analysis (GSVA) Enrichment Analysis

We obtained “c2.cp.KEGG.v7.4.symbols.gmt” files from The Molecular Signatures Database (MSigDB, http://software.broadinstitute.org/gsea/msigdb/index.jsp (accessed on 8 October 2022), and we performed the GSVA enrichment analysis using the “GSVA” R package.

### 2.9. Immunohistochemistry (IHC)

The Human Protein Atlas (HPA, https://www.proteinatlas.org, (accessed on 10 October 2022) was used as an internal relational database to perform the gene atlas. For the purpose of comparing the data in the HPA, we chose only proteins with available IHC staining in either healthy cervix or tumor samples.

### 2.10. Cell Culture and Quantitative Reverse Transcription Polymerase Chain Reaction (qRT-PCR)

The human cervical squamous cell carcinoma cell line (SiHa), the human cervical adenocarcinoma cell line (Hela), the normal human cervix cell line (HUCEC), and the human pancreatic cancer cell line (PANC-1) were all acquired from the Shanghai FuHeng Biotechnology Corporation (Shanghai, China). The incubation conditions of the cells and the qRT-PCR were as described in our previous study [25]. Sequences of the primers are presented in Appendix A.

### 2.11. Statistical Analysis

R software (R Core Team (2022). R: A language and environment for statistical computing. R Foundation for Statistical Computing, Vienna, Austria. (version 4.1.3)) was applied to process the analyses. All *p*-values of < 0.05, as well as false discovery rates (FDR) of q < 0.05, were regarded as statistically significant. T-tests were applied to analyze the qualitative data via SPSS (IBM Corp. Released 2019. IBM SPSS Statistics for Windows, Version 26.0. Armonk, NY: IBM Corp).

## 3. Results

### 3.1. Screening for Anoikis-Related Genes in Cervical Cancer

We selected 126 ANRGS from Harmonizome as well as Genecards databases. In total, 126 ANRGs were retrieved from the tumor tissues and adjacent healthy tissues (combined with GTEx) (Figure 1A). A volcano plot manifested the differential expressions of the ANRGs. Intriguingly, there were 58 differentially expressed ANRGs that overlapped among the TCGA cohort, which included 21 upregulated genes and 37 downregulated genes (*p* < 0.05 and |logFC| > 1; Figure 1B).

### 3.2. Functional Annotation and PPI Analyses of the ANRGs

The GO pathway enrichment analysis revealed that the genes were largely enriched in gland development, the cell junction assembly, the negative regulation of anoikis, the regulation of anoikis, etc., at the biological process (BP) level; in the cell-substrate junction, focal adhesion, collagen-containing extracellular matrix, etc., at the cellular component (CC) level; and in integrin binding, protein serine/threonine/tyrosine kinase activities, protease binding, etc., at the molecular function (MF) level (Figure 2A,B). The KEGG pathway enrichment analysis manifested that these genes were primarily enriched in the PI3K-Akt signaling pathway, focal adhesion and human papillomavirus infection, etc. (Figure 2C). The GSEA identified a set of genes associated with epithelial–mesenchymal transition (Figure 2D). Simultaneously, 58 differential expressions of ANRGs with significant connectivity were utilized to construct a PPI network (Figure 2E,F; Appendix A).

### 3.3. Establishment and Verification of an ANRG Prognostic Signature

The forest plot exhibited 12 ANRGs acquired from the univariate Cox regression analysis (Figure 3A). Nine ANRGs obtained from the LASSO analysis were utilized to investigate the clinical benefit of the ANRGs (Figure 3B,C). Subsequently, by applying a multivariate Cox regression analysis, we extracted six genes (HK2, ITGA5, ROCK1, TP53, IKZF3, and ITGA8), which were independently correlated with the OS among the CESC patients, and we visualized the distribution of the risk scores and the OS status (Figure 3D). We downloaded the gene expression, as well as the clinical profiles in the GSE44001 cohort, from the GEO portal with the aim of validating the precision of the 6-ANRG prognostic signature. We divided 300 samples into high-risk groups (*n* = 153) and low-risk groups (*n* = 147) in accordance with the risk scores applied in the TCGA-CESC cohort. Similarly, this model could effectively predict the OS in the GSE44001 cohort (Figure 3E). The low expression of HK2, ITGA5, and ROCK1 was correlated with good outcomes, whereas the low expression of TP53, IKZF3, and ITGA8 was correlated with poor outcomes. The correlation coefficients are shown in Table 1. Our prognostic signature constructed via the six ANRGs was as follows: risk scores = 0.1502 × HK2 + 0.1744 × ITGA5 + 0.2458 × ROCK1 + (−0.2120) × TP53 + (−0.2876) × IKZF3 + (^−^0.0480) × ITGA8.

The M-curve showed that the patients in the low-risk group could obtain a favorable outcome (*p* < 0.001; Figure 3F). As shown in Figure 3G, the areas under the curve (AUC) of this model in the TCGA-CESC cohort at years 1, 3, and 5 were 0.776, 0.732, and 0.735, respectively. Additionally, the results of the PCA were presented in Appendix A.

### 3.4. Construction of a Prognostic Nomogram for CESC Patients

Referring to the impact of the clinicopathological features on this predictive signature, the nomogram was established by combining the clinical information with the risk scores (Figure 4A). The nomogram was also validated in a calibration plot (Figure 4B). To explore whether the risk scores could be used as an independent prognostic indicator, we carried out a multivariate Cox regression analysis by using the patient’s clinical features, which included age, TNM status, grade, BMI, and risk scores. The findings showed that the risk scores could serve as a sturdy and independent indicator for estimating a patient’s prognosis (Figure 4C; HR  =  2.970, 95% CI 1.897–4.65, *p*  <  0.001).

Our findings revealed that the nomogram with the risk scores underlying the ANRGs was a rewarding approach for predicting the outcomes in clinical practice for CESC patients. Meanwhile, the results of the multi-index ROC curve analysis, combined with the patients’ clinical features, depicted that the AUC area of the risk scores was superior to the nomogram, age, stage, and grade, and it was 0.738 (Figure 4D). These findings demonstrated that the risk-scoring signature had promising prediction capability.

### 3.5. Relationship between Risk Model Constructed by 6-ANRG and Immune Cells

We identified that the risk scores varied significantly in the immune subtypes, and the C1 group showed higher scores compared with the C2 group (Figure 5A). It is well-accepted that TME (tumor microenvironment) contributes greatly to the initiation and progression of tumorigenesis, as well as to immunotherapy; thus, here, we further examined the TME landscape of the CESC patients. Figure 5B showed the ratio of the 22 different infiltrating immune cells in the different samples of the TCGA-CESC cohort (Figure 5B). As shown in the box plot, naive B cells, CD8 T cells, resting memory CD4 T cells, follicular helper T cells, regulatory (Tregs) T cells, gamma delta T cells, M0 macrophages, M2 macrophages, resting mast cells, and activated mast cells were considerably differentially expressed in the low-risk and high-risk groups (Figure 5C).

### 3.6. Potential Therapeutic Value of the 6-ANRG Signature

With the aim of identifying the effects of the risk scores in response to pharmacotherapy, we assessed the relevance between the risk scores and the reaction to medicine in the tumor cell lines. As a result, 34 significantly associated drugs were extracted and compared between the risk scores and the drug sensitivity from the GDSC portal (Figure 6A,B). Among them, four pairs displayed drug sensitivity that was positively associated with the risk scores, including Cytarabine (R = −0.25, *p* = 2.5 × 10^−5^), Bleomycin (R = −0.21, *p* = 0.00049), FTI-277 (R = −0.24, *p* = 5.4 × 10^−5^), and AMG−706 (R = −0.21, *p* = 0.00049), whereas 30 pairs displayed drug sensitivity that was negatively associated with the risk scores, including Zibotentan (R = 0.3, *p* = 3.6 × 10^−7^), Tubastatin A (R = 0.25, *p* = 2.4 × 10^−5^), and Phenformin (R = 0.39, *p* = 4.6 × 10^−11^). Together, the risk scores could serve as a positive indicator for building suitable pharmacotherapy strategies.

### 3.7. GSVA Enrichment Analysis with Different Risk Scores

The 6-ANRG model applied to establish the risk scores had different expression levels between the low-risk and high-risk groups, which was intimately related to the multiple KEGG pathways (Figure 7).

### 3.8. Validation of 6-ANRG for CESC

In brief, some detected proteins exhibited a tendency towards differential expression in tumor samples compared with healthy samples via the available IHC pictures, although sample variability and a picture quality discrepancy were found (Figure 8A). Based on the foregoing bioinformatic analysis, we performed qRT-PCR on the different cell lines for the purpose of validating the expression of these six ANRGs. The trials revealed that HK2, ITGA5, and ROCK1 were positively expressed in the tumor (CESC and PANC-1) cell lines compared with the normal (HUCEC) cell lines (Figure 8B–D). However, TP53, IKZF3, and ITGA8 were considerably downregulated in the tumor cell lines compared with the normal cell lines (Figure 8E–G).

## 4. Discussion

Cervical cancer remains a severe health problem, with approximately half a million females developing the disorder per year worldwide [26]. In some affluent countries, the incidence of cervical cancer is decreasing among women since they have possessed and organized screening and vaccination programs that have decreased their cervical cancer rates [27]. Unfortunately, for a host of developing countries, the shortage in infrastructure and resources has limited the development of such preventative and therapy programs (or they are entirely absent), and so cervical cancer remains the third most pervasive cancer worldwide among women, and 85% of cases occur in developing countries [28]. Due to the limitations and deficiencies in efficient early screening, cervical cancer has often developed to an advanced stage when it is diagnosed [29]. Many women with local tumors continue to receive diverse combinations of surgery, chemotherapy, and radiotherapy in spite of the unresolved concerns about the effective rates of these methods compared to definitive radiotherapy or radical surgery. At the same time, therapies for recurrent CESC remain ineffectual, to a large degree. The preliminary progress in innovative immunotherapeutic methods for treating CESC has displayed propitious outcomes thus far.

Anoikis, a programmed cell death that occurs after cell separation from the proper extracellular matrix, destroys integrin ligation and prevents dysplastic cell growth or attachment to an inappropriate matrix [30]. Recently, based on the discovery that anti-anoikis may cause an epithelial–mesenchymal transition and anchorage-independent growth, both of which are crucial processes in cancer progression and metastasis, anoikis deregulation has captured extensive attention from researchers. Evidence has suggested that anoikis is involved in the mechanism of initiation and progression of tumors in breast cancer [31], head and neck squamous cell carcinoma [32], and hepatocellular carcinoma [33]. However, the direct connection between anoikis and cervical cancer has yet to be fully defined. The existing literature about anoikis has largely focused on the chemical mechanisms of anoikis. Accordingly, several attempts have been made to explore a novel prediction model based on ANRG, which has effective roles in the prognosis of and therapies for cervical cancer.

In this study, firstly, we downloaded the CESC cohort from the TCGA and GTEx databases. Then, we retrieved 126 anoikis-related genes from the Harmonizome and GeneCard databases. Afterward, by integrating these data, we found 58 differentially expressed ANRGs. Subsequently, to further comprehend the functional role of these ANRGs, GO, KEGG, and GSEA enrichment analyses were performed. Consistently, the GO term showed that these genes were interacting with the negative regulation of anoikis, the regulation of anoikis, etc. These findings implied that the differentially expressed ANRGs were significantly associated with tumor development process. With respect to the KEGG pathway analysis, the PI3K-Akt signaling pathway, focal adhesion, human papillomavirus infection, etc., were primarily enriched. The GSEA analysis showed that these ANRGs were enriched in the EMT pathway, which is the leading cause of cancer cell metastasis [34]. Twelve differentially expressed ANRGs were filtered by the univariate COX regression method in the TCGA-CESC dataset. Then, LASSO and multivariate COX regression analyses were utilized to select the six target genes (HK2, ITGA5, ROCK1, TP53, IKZF3, and ITGA8) for establishing a prognosis risk signature for CESC patients. Moreover, we chose an independent GEO dataset (GEO44001) to confirm the prognostic benefit. One of the most-used statistical techniques in clinical research is nomogram-based clinical modeling. A nomogram is helpful for converting complex regression equations into understandable graphic representations, improving the reliability of prediction signature results and risk factors, and other predictor variable probabilities. The age, stage, grade, BMI, and risk score were added to predict the 1-, 3-, and 5-year OS rates for CESC patients, which may facilitate clinical prognostic assessment and therapy. The AUC field of the risk scores was superior to other clinical features. In the present research, after bringing the risk scores into the model, the CESC patients benefited more from the OS rates.

Recently, the tumor microenvironment (TME), as an intrinsic oncogenic mechanism and epigenetic modification, has become a research hotspot [35]. The TME contains a few non-cancerous cells that influence cancer cell survival and has a significant effect on tumor metastasis and drug treatment efficacy [36,37]. By analyzing the ratio of 22 types of immune cells in two risk groups, we found that the infiltration levels of resting memory CD4 T cells, M0 macrophages, M2 macrophages, and activated mast cells were significantly upregulated in the high-risk group, with an adverse OS rate, showing the critical work of these cells in the initiation and progression of CESC. Higher proliferative levels of naive B cells, CD8 T cells, follicular helper T cells, regulatory (Tregs) T cells, gamma delta T cells, and resting mast cells were in the low-risk group. The results of the immune infiltration analysis also indicated that the low-risk group had a positive prognostic capability compared with the high-risk group.

Further, drug sensitivity analysis was applied to validate whether the risk scores could be utilized to predict the chemotherapy sensitivity rates of CESC patients. The modeling data of the drug analysis revealed that the high-risk patients were highly sensitive to treatment with Cycloabine, Bleomycin, FTI-277, and AMG − 706, whereas the patients in the low-risk group were more sensitive to treatment with Zibotentan, Tubastatin A, Phenormin, etc. These findings have great significance for the clinical treatment of CESC patients.

Ultimately, the qRT-PCR validated that HK2, ITGA5, and ROCK1 were upregulated in the CESC cell lines compared with the HUCEC (negative control) cell lines. It is worth noting that these three genes were significantly more highly expressed in the PANC-1 (positive control) cell lines compared with the HUCEC cell lines, which indicated that they were differentially expressed in the different cancer types, in accordance with previous studies [38,39,40]. Consistent with previous studies, the TP53 gene was considerably upregulated in the normal cell lines compared with the tumor cell lines [41]. IKZF3 and ITGA8 were considerably more lowly expressed in the normal cell lines compared with the tumor cell lines. According to our understanding, no previous studies have explored whether IKZF3 and ITGA8 are engaged in the initiation and progression of pancreatic cancer. Hence, we were unable to understand how these two genes exerted their functions in pancreatic cancer. Nonetheless, this did not affect our conclusion, and the results of the qRT- PCR were in line with these foregoing bioinformatic analyses.

Furthermore, in the existing studies, HK2 was shown to be upregulated in multiple types of cancers and correlated with increased aerobic glycolysis [42]. Accumulating confirmations have indicated that HK2 exerts its role more often in glycolysis, as well as in cell survival. In the last two decades, studies have revealed that the overexpression of HK2 has existed in many types of cancer, such as cervical cancer [43], esophageal carcinoma [44], etc. Many studies have argued that ROCK1 (Rho-associated coiled-coil containing protein kinase 1) is engaged in cell proliferation, motility, and metastasis, and so ROCK1 may be a promising therapeutic target for carcinoma therapy [45,46]. Additionally, recent attention has been paid to ROCK1 as a downstream factor of OIP5-AS1 in the regulation of CESC occurrence and development [47]. Prior research has indicated that integrin-alpha-5 (ITGA5) fosters the occurrence of many types of tumors, and its overexpression may be pivotal to tumor cell invasion [48]. ITGA8 (integrin-alpha-8) is involved in the occurrence of several types of cancers [49]. Some researchers have also suggested that ITGA8 was considerably lowly expressed in lung adenocarcinoma (LUAD), and ITGA8 could be a potential prognostic biomarker in LUAD patients [50]. The tumor suppressor gene TP53 served as a DNA binding transcription effector, and it was present at poor levels in normal cells owing to the rapid proteasome-mediated turnover of the MDM2 ubiquitin ligase [51]. Our literature review showed that TP53 exerted its tumor suppressor role by regulating DNA repair, apoptosis, and cell cycle arrest in various types of tumors, including CESC tumors [52]. IKZF3 was the crucial regulator of lymphoid differentiation, and it belongs to the Ikaros family of lymphoid transcription factors [53]. Yang and Lin-Kai et al. showed that low levels of IKZF3 indicated poor clinical outcomes in skin cutaneous melanoma (SKCM) [54]. Recently, studies have explored the relationships between IKZF3 and hypoxia-immunity in cervical cancer, which may be beneficial to the development of novel prognostic biomarkers and therapeutic strategies for cervical cancer [55]. Meanwhile, IKZF3, as the most statistically significant gene, will be further investigated for its molecular mechanisms in subsequent studies and experiments.

We have to point out that cervical squamous cell carcinoma and endocervical adenocarcinoma are two different histologic groups. Since the cervical cancer data provided in the TCGA database include both cervical squamous cell carcinoma and endocervical adenocarcinoma, we were unable to obtain or modify the original data provided in the database. Thus, we downloaded gene expression as well as clinical profiles in the GSE44001 cohort from the GEO portal to validate the precision of the 6-ANRG prognostic signature, and the results were consistent with the TCGA-CESC cohort. Moreover, in subsequent qRT- PCR experiments, the expression levels of six genes in SiHa (The human cervical squamous cell carcinoma cell line) and Hela (the human cervical adenocarcinoma cell line) were consistent with these foregoing bioinformatic analyses. Thus, the error caused by the two histological cervical cancers is within the permissible range and does not affect our experimental results.

Although we achieved some positive and meaningful results in this study, some limitations existed due to the following reasons: First and foremost, as the gene expression profiles adopted in our research were from different portals, this discrepancy may have produced bias in the analysis procedures. Therefore, the sample size should be expanded in future studies to overcome the problem of discrepancies in the databases. Secondly, before the 6-ANRG signature of CESC can be applied in clinical work, we require further in vivo, in vitro, and in population studies to verify our hypothesis based on the molecular mechanism of the 6-ANRG signature and drug predictions. Considering these limitations, it seems that the predictive function of our CESC 6-ANRG signature is worthy of further study.

## 5. Conclusions

Our study is the first to investigate the prognostic function of ANRGs in CESC. The 6-ANRG model was established to predict the prognoses of CESC patients. This model was verified in the GSE44001 cohort, which demonstrated that it has crucial clinical value. This research underlines the pivotal clinical implications of the 6-ANRG signature and will assist in the development of immune therapeutic strategies for CESC patients, though it awaits further clinical experiments.

## Figures and Tables

**Figure 1 medicina-59-00358-f001:**
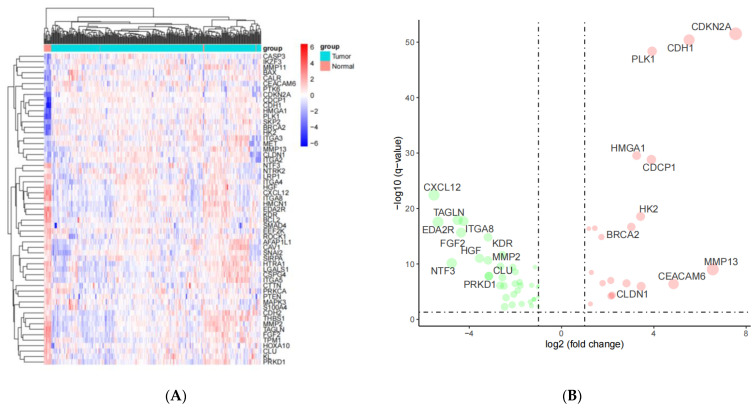
The identified differentially expressed anoikis-related genes (ANRGs). (**A**) Heatmap of the ANRGs in cervical squamous cell carcinoma and endocervical adenocarcinoma (CESC). (**B**) Volcano plot of differentially expressed ANRGs. *p*-values of <0.05 and |logFC| values of >1 were utilized as the filtering criteria.

**Figure 2 medicina-59-00358-f002:**
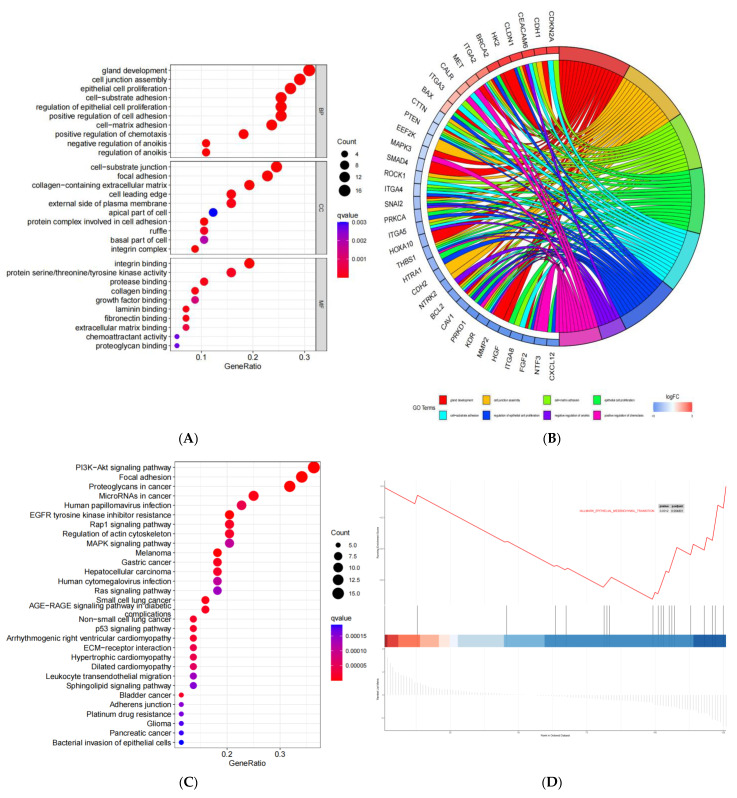
Functional enrichment of the differentially expressed ANRGs. (**A**) A bubble plot of the Gene Ontology (GO) enrichment analysis. (**B**) A circle plot shows the important signal pathways that were correlated with these genes. (**C**) A Kyoto Encyclopedia of Genes and Genomes (KEGG) signal pathway enrichment analysis. (**D**) The genes that were identified through the Gene Set Enrichment Analysis (GSEA). (**E**,**F**) Protein–Protein Interaction (PPI) analysis of the differentially expressed ANRGs.

**Figure 3 medicina-59-00358-f003:**
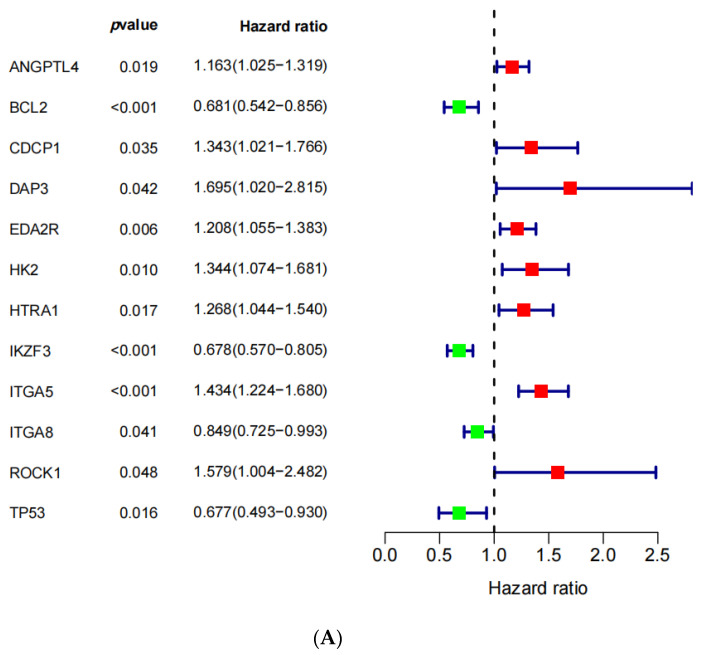
The identified anoikis-related prognosis signature. (**A**) The univariate Cox regression analysis of the differentially expressed ANRGs. (**B**) The least absolute shrinkage and selection operator (LASSO) analysis of the 12 prognostic ANRGs. (**C**) The coefficient profile plot of the 12 prognostic ANRGs. (**D**,**E**) The relationships between the risk scores (top), the risk scores and the survival status (middle), the risk scores and the gene expression in the TCGA-CESC training set, and the risk scores and the GSE44001 validation set. (**F**) The Kaplan–Meier (K-M) curve shows the prognosis for the different risk groups. (**G**) The time-dependent receiver operational feature curve (ROC) curve for the OS at years 1, 3, and 5. AUC: the areas under the curve.

**Figure 4 medicina-59-00358-f004:**
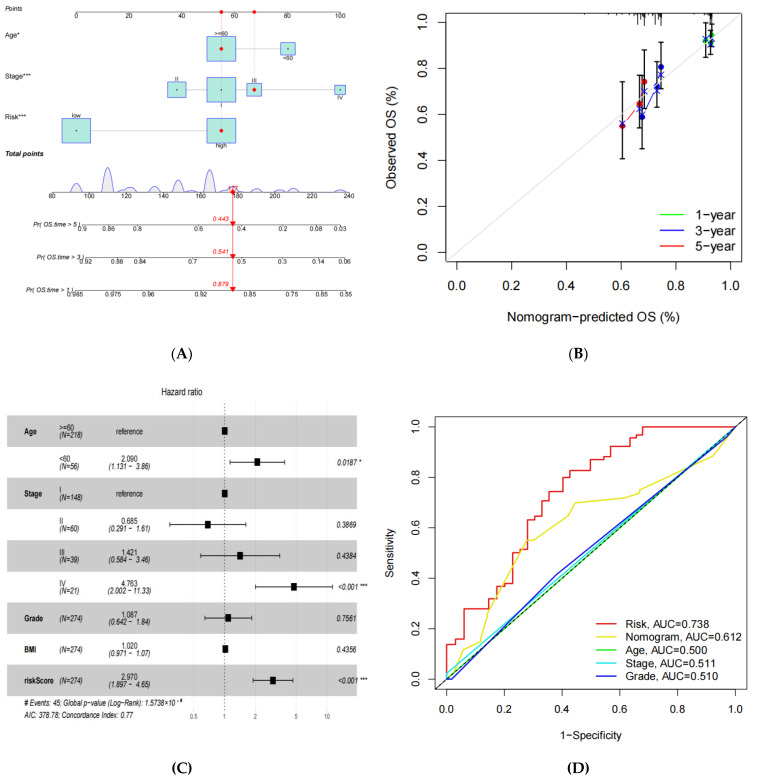
Establishing a nomogram for CESC to predict overall survival rates. (**A**) The nomogram is based on clinicopathological factors and the risk scores. (**B**) Calibration plot of the prognostic model for overall survival (OS) years 1, 3, and 5. (**C**) Multivariate Cox regression model analysis, which included the clinical traits and the risk scores in the TCGA-CESC cohort. (**D**) Multi-index ROC curve analysis of the risk scores and clinicopathological factors. * *p* < 0.05 and *** *p* < 0.001.

**Figure 5 medicina-59-00358-f005:**
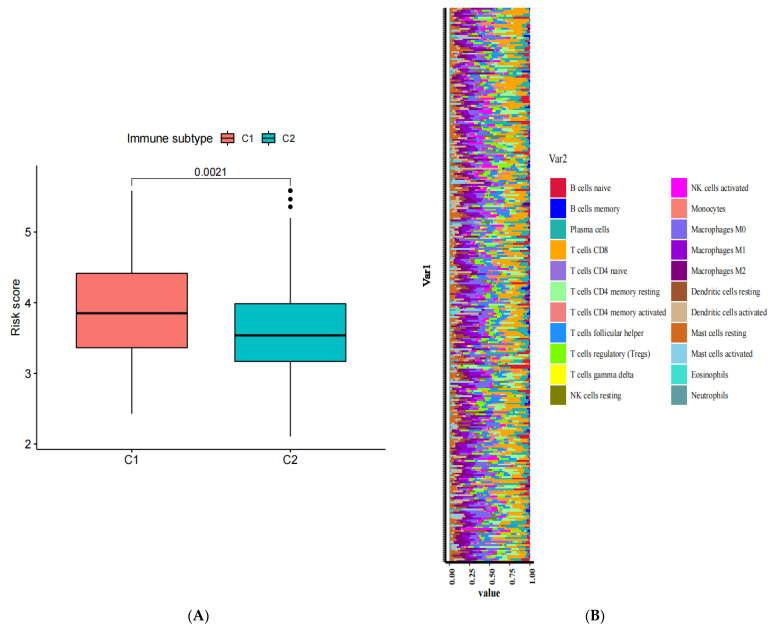
The relationships between the (6-anoikis-related gene) 6-ANRG signature and the immune cells. (**A**) The box plot shows the correlation between the risk scores and the immune subtypes. (**B**) The relative proportion of the infiltrating immune cells in CESC patients from the TCGA cohort. (**C**) The box plot shows the expression patterns in the 22 immune cells from the different risk groups. * *p* < 0.05, ** *p* < 0.01, and *** *p* < 0.001.

**Figure 6 medicina-59-00358-f006:**
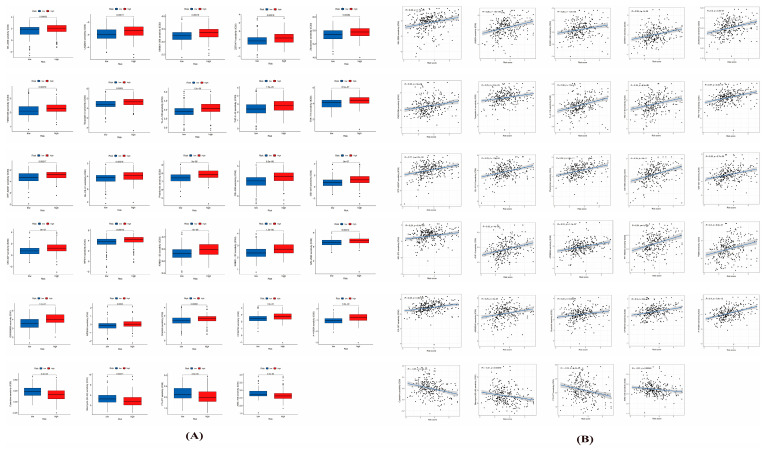
The relationships between the risk scores and drug sensitivity. (**A**) The box plot depicts the drug sensitivity for the different risk groups. (**B**) The correlation between the risk scores and drug sensitivity was evaluated by Spearman analysis.

**Figure 7 medicina-59-00358-f007:**
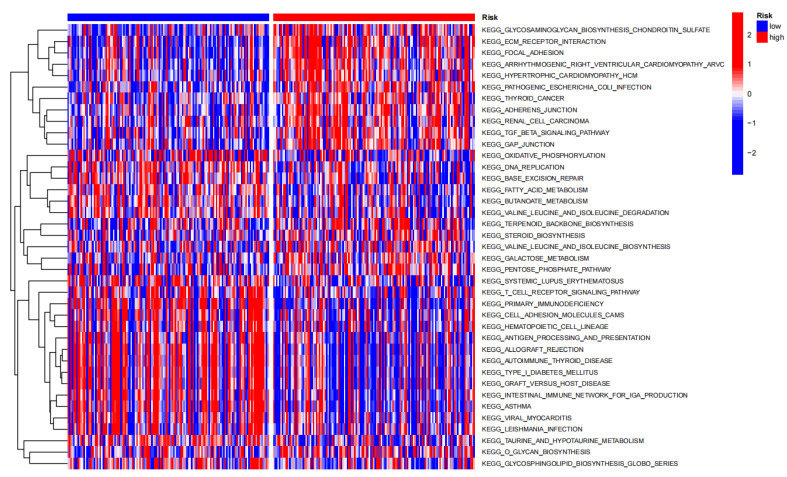
GSVA enrichment analysis for the different risk groups. The heatmap depicts the expression profiles of the KEGG pathways between the high-risk group and the low-risk group.

**Figure 8 medicina-59-00358-f008:**
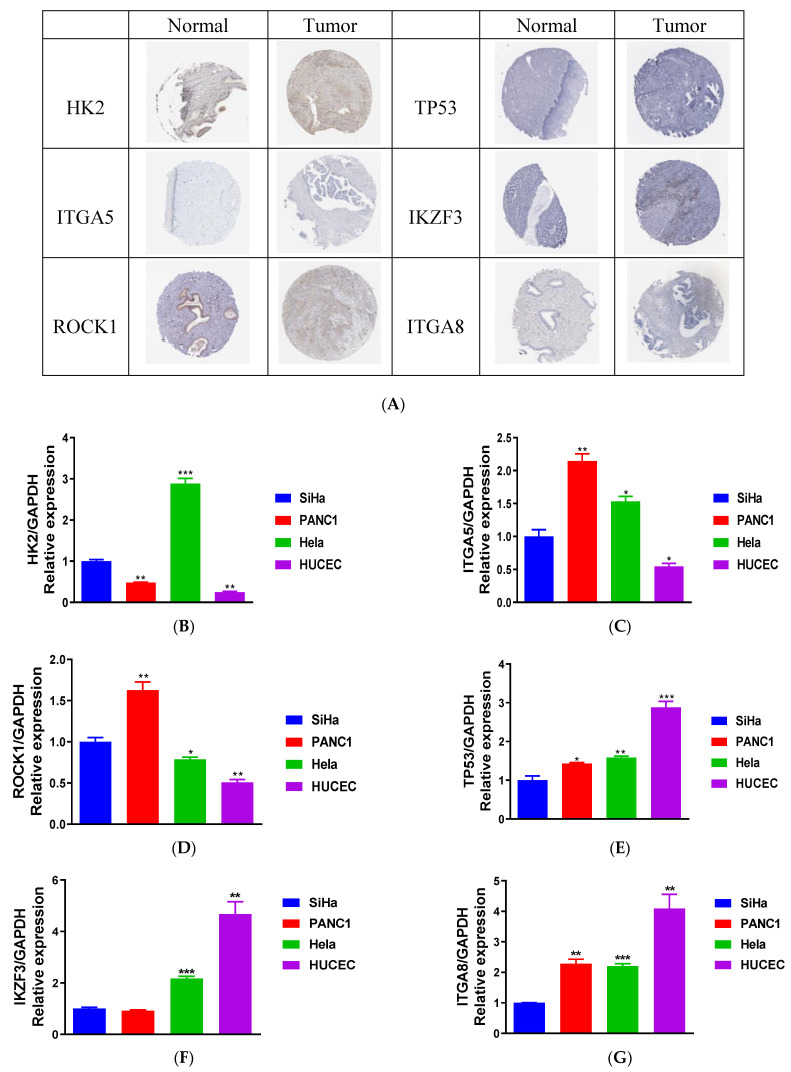
Immunohistochemistry (IHC) and the expression of the 6-ANRG model in the cervical cancer cell lines. (**A**) The IHC pictures derived from the Human Protein Atlas (HPA) database show trends towards differential expression in both the tumor and normal samples. (**B**–**G**) The expression levels of the six anoikis-related genes in the Hela and SiHa cell lines compared to the PANC-1 (positive control) and HUCEC (negative control) cell lines. SiHa: the human cervical squamous cell carcinoma cell line; Hela: the human cervical adenocarcinoma cell line; HUCEC: the normal human cervix cell line; PANC-1: the human pancreatic cancer cell line. * *p* < 0.05, ** *p* < 0.01, and *** *p* < 0.001.

**Table 1 medicina-59-00358-t001:** Multivariate Cox regression analysis for the prognostic signature based on the six ANRGs.

ID	Coefficient	HR	HR 95L	HR 95H	*p*-Value
HK2	0.15021	1.16208	0.9019	1.4973	0.24543
ITGA5	0.17438	1.19051	0.9926	1.4278	0.06007
ROCK1	0.24578	1.27862	0.7658	2.1349	0.34739
TP53	−0.21204	0.80893	0.5681	1.1518	0.23954
IKZF3	−0.28758	0.75008	0.6137	0.9168	0.00498
ITGA8	−0.04797	0.95316	0.7738	1.1741	0.65198

ID: Identity document HR: Hazard ratio.

## Data Availability

The data used and/or analyzed in the present research are available from the corresponding author on reasonable request.

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
