# Peer review of "Identification and Validation of the Anoikis-Related Gene Signature as a Novel Prognostic Model for Cervical Squamous Cell Carcinoma, Endocervical Adenocarcinoma, and Revelation Immune Infiltration"

_medicina, 2023, doi:10.3390/medicina59020358_

Round 1

Reviewer 1 Report

The authors present their results of their analysis of anoiksis-related genes in cervical squamous cell carcinoma and development of a predictive model. Overall, their research question is important and novel, and their research approach seems appropriate. Their study should be considered hypothesis-generating.

My general comments:

1)   The manuscript will need extensive editing for English language and grammar if accepted

2) Lines 327-329 discuss pancreatic cancer, not cervical cancer.

3) The only ANRG that was statistically prognostic was IKZF3 on multivariate Cox regression analysis.  The discussion should pay more attention to this ANRG, as well as discuss the impact on their risk nonogram.

4) Along the same lines, the discussion failed to adequately discuss the nomogram and its potential use in translational medicine.

5) Squamous and glandular cervical malignancies are different.  There were no data presented and there was no discussion of the difference between these two histologic groups.

6) Stating "etc." in the manuscript is not acceptable (used often).

Author Response

Response to Reviewer 1 Comments

Thank you very much for your email on 23 Jan 2023 with which you sent us the reviewer ' s report on our paper with the Manuscript ID: medicina-2108773. We also wish to take this opportunity to thank the reviewer for his constructive comments and valuable recommendations. We have carefully revised the manuscript according to reviewer’s suggestion.

Our responses to several comments are listed below:

Point 1: The manuscript will need extensive editing for English language and grammar if accepted.

Response 1: This manuscript has been revised extensively according to the reviewers constructive suggestions. In addition, the language expression of the manuscript has been improved with the help of language editing service of MDPI’s journals.

Point 2: Lines 327-329 discuss pancreatic cancer, not cervical cancer.

Response 2: We found the following sentence to Lines 327-329 : “Evidences suggested that anoikis involvement in the mechanism of initiation and progression of tumor in breast cancer, head and neck squamous cell carcinoma and hepatocellular carcinoma.", in fact, this sentence did not discuss pancreatic cancer, we just demonstrated that anoikis involvement in the mechanism of initiation and progression of various tumor.

If you meant the sentences in paragraph 6 of our discussion, we indeed discussed pancreatic cancer, because we chose the human pancreatic cancer cell line (PANC-1) as a positive control and meanwhile the qRT-PCR result of PANC-1 cell line did not affect our conclusion.

Point 3: The only ANRG that was statistically prognostic was IKZF3 on multivariate Cox regression analysis. The discussion should pay more attention to this ANRG, as well as discuss the impact on their risk nomogram.

Response 3: Thanks for your kind suggestions, which is valuable for improving the accuracy of the manuscript. More discussions have been included in the revised manuscript (Line 438-443). 

Point 4: Along the same lines, the discussion failed to adequately discuss the nomogram and its potential use in translational medicine.

Response 4: Thanks for your comments, which is highly appreciated. More discussions have been included in the revised manuscript (Line 375-381). As far as translational medicine is concerned, before 6-ANRG signature of CESC can really be applied in clinical work, we need to go further in vivo, in vitro, and in population studies to verify our study. And the above-mentioned sentences have been included in paragraph 8 in the discussion.

Point 5: Squamous and glandular cervical malignancies are different. There were no data presented and there was no discussion of the difference between these two histologic groups.

Response 5: We agree with the reviewer on this point. Since the cervical cancer data provided in the TCGA database includes both squamous and glandular cervical malignancies, we were unable to obtain or modify the original data provided in the database. So, we downloaded gene expression as well as clinical profiles in GSE44001 cohort from GEO portal to validate the precision of the 6-ANRG prognostic signature and the results was consistent with TCGA-CESC cohort. Moreover, in subsequent qRT- PCR experiments, the expression level of six genes in SiHa(The human cervical squamous cell carcinoma cell line) and Hela(the human cervical adenocarcinoma cell line) were consistent with these foregoing bioinformatic analyses. Thus, the error caused by the two histological cervical cancers is within the permissible range and does not affect our experimental results. For these reasons, we chose not to make this change.

Point 6: Stating "etc." in the manuscript is not acceptable (used often).

Response 6: We agree with the comment and have replaced the word in the revised manuscript. And the English editing service provided by MDPI Press has also helped us improve this problem.

Finally, we sincerely hope that this revised manuscript could address most of your comments and suggestions. We really appreciated for your warm work earnestly and hope that the revision will meet with approval. Once again, thank you very much for your comments and suggestions.

Reviewer 2 Report

Dear Authors 

I read the paper: "Identification and validation of anoikis-related gene signature as a novel prognostic model for cervical squamous cell carcinoma and endocervical adenocarcinoma and revelation immune infiltration", which falls whithin the aim of Medicina. Honestly, the topic is interesting enough to attract the readers' attention, but the article would benefit from a minor revision. I have specific recommendations that I describe under each section:

Abstract

1)LINE 17: "Cervical squamous cell carcinoma and endocervical adenocarcinoma (CESC) are still malignant disorders with adverse prognosis for advanced patients. Anoikis involved in tumor metastasis, which facilitated the survival and separation of tumor cells from the initial site. Unfortunately, rarely studied in literature has attended to the prognosis character of anoikis for patients with CESC"

The Background and Objectives section should be concise and precise, and this paragraph is repetitious. I suggest rewriting this line.

2)LINE 21: Materials and Methods. When and where was the study conducted?

Introduction

3) Line 45: "CESC" is an abbreviation. The authors must add the complete name prior to the acronym "Cervical squamous cell carcinoma and endocervical adenomacinoma"

4)Line 48:"The overall survival (OS) is at approximately 70% five years after pharmacotherapy with locally advanced cervical cancer". Because the study appeals to an international audience, cervical squamous cell carcinoma management should be well reported, citing the following papers:

  • https://doi.org/10.1016/S0025-6196(11)61104-X
  • https://doi.org/10.1016/j.ygyno.2022.04.010
  • DOI: 10.1596/978-1-4648-0349-9_ch4

Materials and Methods

5) When and where did the study take place? Do medical files contain demographic and clinical information? Could the authors provide any additional details to better comprehend the clinical situation?

6) Inclusion/exclusion criteria should be better clarified.

7) Proof-reading by a native English speaker is mandatory in order to improve readability and correct several typos.

In conclusion, the paper is good and appeared original with substantial importance, but more studies are needed to support the study's conclusion.

I suggest a minor revision.

Author Response

Response to Reviewer 2 Comments

Thank you very much for your e-mail dated 27 Jan. 2023 regarding the review results of our Manuscript ID: medicina-2108773). This manuscript has been revised very carefully according to the constructive comments from the reviewers. Their main concerns have been well addressed. 

The following is the highlighted revisions that we have made:

Point 1: Line 17: "Cervical squamous cell carcinoma and endocervical adenocarcinoma (CESC) are still malignant disorders with adverse prognosis for advanced patients. Anoikis involved in tumor metastasis, which facilitated the survival and separation of tumor cells from the initial site. Unfortunately, rarely studied in literature has attended to the prognosis character of anoikis for patients with CESC"The Background and Objectives section should be concise and precise, and this paragraph is repetitious. I suggest rewriting this line.

Response 1: Thanks for your comments, which is highly appreciated. Since CESC and anoikis are the focus of the discussion in this paper, and little attention was paid to to the prognosis character of anoikis for patients with CESC, which served as one of the innovative points of this paper. So, we think these few sentences in the Abstract are necessary. Thanks again to the reviewer on suggesting to further improve this manuscript, we have studied comments carefully and have made corresponding corrections which we hope meet with approval.

Point 2: Line 21: Materials and Methods. When and where was the study conducted?

Response 2: Thank you for your comment, since this study is based on data analysis from a number of public databases such as TCGA, and no additional information about the time and place of the study is required.

Point 3: Line 45: "CESC" is an abbreviation. The authors must add the complete name prior to the acronym "Cervical squamous cell carcinoma and endocervical adenomacinoma"

Response 3: Thanks for your comments, the complete name of "CESC" has been written in Line 46-47.

Point 4: Line 48:"The overall survival (OS) is at approximately 70% five years after pharmacotherapy with locally advanced cervical cancer". Because the study appeals to an international audience, cervical squamous cell carcinoma management should be well reported, citing the following papers:

https://doi.org/10.1016/S0025-6196(11)61104-X

https://doi.org/10.1016/j.ygyno.2022.04.010

DOI: 10.1596/978-1-4648-0349-9_ch4

Response 4: Thanks for the references, which are now included in the revised manuscript. Specific references are listed as follows:

  1. Long III, Harry J., Nadia NI Laack, and Bobbie S. Gostout. "Prevention, diagnosis, and treatment of cervical cancer." In Mayo Clinic Proceedings, 82 2007:(1566-1574). doi.org/10.1016/S0025-6196(11)61104-X.
  2. Ronsini, Carlo, et al. "Laparo-assisted vaginal radical hysterectomy as a safe option for Minimal Invasive Surgery in early stage cervical cancer: A systematic review and meta-analysis." Gynecologic oncology (2022). doi.org/10.1016/j.ygyno.2022.04.010.
  3. Denny, Lynette, et al. "Cervical cancer." Cancer: Disease Control Priorities, Third Edition (Volume 3) (2015). doi: 10.1596/978-1-4648-0349-9_ch4.

Point 5: When and where did the study take place? Do medical files contain demographic and clinical information? Could the authors provide any additional details to better comprehend the clinical situation?

Response 5: We deeply appreciate your suggestion. Since this study is based on data analysis from a number of public databases such as TCGA, and no additional information about the time and place of the study is required. And the medical files contain clinical information such as age, TNM status, grade and BMI, without demographic information. All the clinical profiles could be obtained from The Cancer Genome Atlas (TCGA, https://portal.gdc.cancer.gov), the Genotype-Tissue Expression (GTEx, http://xena.ucsc.edu/) and the Gene Expression Omnibus (GEO, https://www.ncbi.nlm.nih.gov/geo/) databases.

Point 6: Inclusion/exclusion criteria should be better clarified.

Response 6: Thanks for your comments, all data in this study were obtained from publicly available web-based databases, so no inclusion and exclusion criteria were required.

Point 7: Proof-reading by a native English speaker is mandatory in order to improve readability and correct several typos.

Response 7: Thanks for your constructive suggestion, which is highly appreciated. With the help of language editing service of MDPI’s journals, we have carefully scrutinized the manuscript, and made corresponding revisions including some typos, grammatical errors and long sentences, and so on.

Finally, we would like to thank you again for taking the time to review our revision.

Round 2

Reviewer 1 Report

Thank you for the opportunity to review the revised manuscript.  Overall, the authors have adequately addressed my concerns.  However, I remained concerned about the lack of discussion of the difference between squamous and adenocarcinoma in the revised manuscript.  The authors' response to this query was well-written (see below).  I suggest they add their response to the manuscript; this would adequately clarify the issue.

Point 5: Squamous and glandular cervical malignancies are different. There were no data presented and there was no discussion of the difference between these two histologic groups.

Response 5: We agree with the reviewer on this point. Since the cervical cancer data provided in the TCGA database includes both squamous and glandular cervical malignancies, we were unable to obtain or modify the original data provided in the database. So, we downloaded gene expression as well as clinical profiles in GSE44001 cohort from GEO portal to validate the precision of the 6-ANRG prognostic signature and the results was consistent with TCGA-CESC cohort. Moreover, in subsequent qRT- PCR experiments, the expression level of six genes in SiHa(The human cervical squamous cell carcinoma cell line) and Hela(the human cervical adenocarcinoma cell line) were consistent with these foregoing bioinformatic analyses. Thus, the error caused by the two histological cervical cancers is within the permissible range and does not affect our experimental results. For these reasons, we chose not to make this change.

Author Response

Response to Reviewer 1 Comments

Thank you very much for your email on 02 Feb 2023 with which you sent us the reviewer ' s report on our paper with the Manuscript ID: medicina-2108773. We also wish to take this opportunity to thank the reviewer for his constructive comments and valuable recommendations. We have carefully revised the manuscript according to reviewer’s suggestion.

Our responses to several comments are listed below:

Point 5: Squamous and glandular cervical malignancies are different. There were no data presented and there was no discussion of the difference between these two histologic groups. 

Thank you for the opportunity to review the revised manuscript. Overall, the authors have adequately addressed my concerns. However, I remained concerned about the lack of discussion of the difference between squamous and adenocarcinoma in the revised manuscript. The authors' response to this query was well-written (see below).  I suggest they add their response to the manuscript; this would adequately clarify the issue.

“We agree with the reviewer on this point. Since the cervical cancer data provided in the TCGA database includes both squamous and glandular cervical malignancies, we were unable to obtain or modify the original data provided in the database. So, we downloaded gene expression as well as clinical profiles in GSE44001 cohort from GEO portal to validate the precision of the 6-ANRG prognostic signature and the results was consistent with TCGA-CESC cohort. Moreover, in subsequent qRT- PCR experiments, the expression level of six genes in SiHa(The human cervical squamous cell carcinoma cell line) and Hela(the human cervical adenocarcinoma cell line) were consistent with these foregoing bioinformatic analyses. Thus, the error caused by the two histological cervical cancers is within the permissible range and does not affect our experimental results. For these reasons, we chose not to make this change.”

Response 5: Thanks for your kind suggestions, which is valuable for improving the accuracy of the manuscript. The discussions have been included in the revised manuscript (Line 444-455). 

Finally, we would like to thank you again for taking the time to review our revision.
